# Response of Anthocyanin Accumulation in Pepper (*Capsicum annuum*) Fruit to Light Days

**DOI:** 10.3390/ijms23158357

**Published:** 2022-07-29

**Authors:** Yan Zhou, Muhammad Ali Mumtaz, Yonghao Zhang, Huangying Shu, Yuanyuan Hao, Xu Lu, Shanhan Cheng, Guopeng Zhu, Zhiwei Wang

**Affiliations:** 1Key Laboratory for Quality Regulation of Tropical Horticultural Crops of Hainan Province, School of Horticulture, Hainan University, Haikou 570100, China; 17071001110007@hainanu.edu.cn (Y.Z.); ali_maken96@icloud.com (M.A.M.); hnhyshu@163.com (H.S.); yyhao630@163.com (Y.H.); luxu@hainanu.edu.cn (X.L.); 990865@hainanu.edu.cn (S.C.); guopengzhu@163.com (G.Z.); 2Sanya Nanfan Research Institute of Hainan University, Hainan Yazhou Bay Seed Laboratory, Sanya 572025, China; 3Institute of Tropical Horticulture Research in Hainan Academy of Agricultural Sciences, Haikou 571100, China; m17886707503@163.com

**Keywords:** purple pepper, anthocyanin, light days, WGCNA

## Abstract

Light is the key factor affecting the synthesis of anthocyanins in pepper. In this study, pepper fruit under different light days was used as experimental material to explore the synthesis of anthocyanins in purple pepper. A total of 38 flavonoid metabolites were identified in the purple pepper germplasm HNUCA21 by liquid chromatography–tandem mass spectrometry (LC-MS/MS), of which 30 belong to anthocyanins. The detected anthocyanin with the highest content was Delphinidin-3-O-glucoside (17.13 µg/g), which reached the maximum after 168 h of light treatment. Through weighted gene co-expression network analysis (WGCNA), the brown module was identified to be related to the early synthesis of anthocyanins. This module contains many structural genes related to flavonoid synthesis, including *chalcone synthase* (*CHS* 107871256, 107864266), *chalcone isomerase* (*CHI* 107871144, 107852750), *dihydroflavonol 4-reductase* (*DFR* 107860031), *flavonoid 3′ 5′-hydroxylase* (*F3’5’H* 107848667), *flavonoid 3′-monooxygenase* (*F3M* 107862334), *leucoanthocyanidin dioxygenase* (*LDOX* 107866341), and *trans-cinnamate 4-monooxygenase* (*TCM* 107875406, 107875407). The module also contained some genes related to anthocyanin transport function, such as *glutathione S-transferase* (*GST* 107861273), *anthocyanidin 3-O-glucosyltransferase* (*UDPGT* 107861697, 107843659), and *MATE* (107863234, 107844661), as well as some transcription factors, such as *EGL1* (107865400), *basic helix-loop-helix 104* (*bHLH104* 107864591), and *WRKY44* (107843538, 107843524). The co-expression regulatory network indicated the involvement of *CHS*, *DFR*, *CHI*, and *EGL1*, as well as two *MATE* and two *WRKY**44* genes in anthocyanin synthesis. The identified genes involved in early, middle, and late light response provided a reference for the further analysis of the regulatory mechanism of anthocyanin biosynthesis in pepper.

## 1. Introduction

Light is an important factor affecting the formation of anthocyanins. Different light qualities and light intensities affect plant anthocyanin synthesis and metabolism [1]. The anthocyanin synthesis of purple pepper is induced by light. Different light days induce the differential expression of several structural genes and transcription factors involved in anthocyanin synthesis, resulting in changes in anthocyanin accumulation [2].

Previous studies have found key roles of transcription factors in the synthesis of anthocyanins. *WRKY44* plays a role in regulating the promoters of *F3’H* and *F3’5’H* genes during kiwifruit coloring, and *F3’H* and *F3’5’H* are the key branches of different anthocyanin synthesis types [3]. Catechin synthesis depends on the flavonoid pathway, and *WRKY44* plays a key role in the regulation of catechin synthesis in the callus of purple-black kiwifruit [3].

Similarly, bHLH transcription factors, which can independently regulate CHS, DFR, and UFGT, are the key transcriptional regulators of anthocyanin biosynthesis [4].

The transport and accumulation of anthocyanins in vacuoles are mainly mediated by glutathione S-transferases (GSTs) and MATE transporters [5,6]. GSTs are a superfamily of multifunctional enzymes that catalyze the binding of glutathione (GSH, glutamylcysteine) to electrophilic sites [7,8]. At present, three hypotheses of glutathione S-transferase (GST), membrane transporter, and vesicle transport have been identified as related to anthocyanin transport [9]. The first mechanism involves GST sequestering anthocyanins from the cytoplasm into vacuoles, the second relies on transporters located in the vacuolar membrane, including ATP binding cassette (ABC) proteins and transporters, and the last mechanism is based on the fusion of the smaller vesicular structures containing cytoplasmic anthocyanin bodies with central vacuoles [10,11,12,13]. An indirect function of GSTs is to remove reactive oxygen species (ROS), including superoxide radicals, hydroxyl radicals, alkoxy groups, hydroperoxides, and singlet oxygen, which are constantly produced in the mitochondria [14,15,16,17]. Once anthocyanins are synthesized, GST and transporters accumulate in vacuoles to prevent anthocyanins from being oxidized [18,19,20].

Pepper fruit grows to its maximum size about 21 days after pollination and the purple color is prominent throughout the growth stage [21,22]. Anthocyanins are continuously synthesized throughout the development period of immature fruit of pepper, but which time point is the key period of anthocyanin synthesis and which gene plays a key role in this process still need to be elucidated [23,24]. By constructing a weighted correlation network analysis, we sought to identify the genes involved in early, middle, and late light response, and further analyzed the differential regulation of structural genes involved in anthocyanin biosynthesis [25,26,27,28]. Our work identified the sequentially expressed light response genes, which will help us further clarify the mechanism of light-induced capsicum pericarp coloring and enrich our understanding of the regulation of anthocyanin biosynthesis [29,30,31].

## 2. Results

### 2.1. Total Anthocyanin, Chlorophyll, and Carotenoid Concentrations 

The rich color of pepper is mainly based on anthocyanins, carotenoids, chlorophyll, and other pigment substances. In the young fruit stage, purple pepper is rich in anthocyanins and chlorophyll, whereas when the fruit is mature, it contains an abundance of carotenoids. After bagging the pepper for two weeks, half of the fruit was treated with light (Figure 1). After removing the bag for 48 h, the peel color of the pepper changed from light purple (Figure 1a) to a purple-red color (Figure 1b) and finally to a purple-black color (Figure 1c). The fruit color on the shaded surface remained yellow and white (Figure 1d–f). The total anthocyanin content of pepper increased rapidly from 6 µg/g in 48 h to 46.3 µg/g in 168 h after removing the bag. The anthocyanin content of the shaded surface did not change significantly at the three-time points and remained between 0.9 and 1.8 µg/g (Figure 2a). The chlorophyll content of light treatment also increased gradually with the extension of light time, from 5 µg/g in 48 h to 43 µg/g in 168 h, and the chlorophyll content of the shaded surface remained at a low level (Figure 2b). The total amount of carotenoids did not change significantly before and after illumination (Figure 2c).

### 2.2. Analysis of Flavonoids in Pepper Peel Samples by Liquid Chromatography–Tandem Mass Spectrometry (LC-MS/MS)

The peel samples at 48 h, 72 h, and 168 h after bagging and their corresponding shaded surface peel samples were analyzed by liquid chromatography–tandem mass spectrometry. According to the type and content histogram of flavonoids in each sample (Figure 3), 38 kinds of flavonoid metabolites were detected in purple pepper, including 30 kinds of anthocyanins. The seven flavonoids with the highest content in the sample were Quercetin-3-O-glucoside, Rutin, Naringenin-7-O-glucoside, Kaempferol-3-O-rutinoside, Delphinium-3-O-glucoside, Delphinium-3-O-rutin, and Petunia-3-O-(coumarin)-glu. The specific contents of all metabolites in the peel samples with different treatments are presented in Appendix A.

### 2.3. Analysis of Transcriptome Data Quality of Pepper Peel Samples

Transcriptomic changes in the pepper peel samples treated with light for 48 h, 72 h, and 168 h and their corresponding shaded surface samples were analyzed by RNA sequencing technology (RNA-seq). The statistical results of transcriptome sequencing of samples treated with different treatments are shown in Table 1. In total, 125.96 G clean bases were obtained from 18 fruit samples of purple pepper, the clean bases of each sample were more than 6.45 G, the percentage of Q30 bases was more than 94.01%, and the GC values were between 41.74% and 42.97%.

### 2.4. Comparative Statistical Analysis of Transcriptome Data of Pepper Peel Samples

Table 2 shows that the number and percentage of reads on the reference genome compared to this transcriptome sequencing are between 33,933,296 (75.08%) and 44,704,508 (96.09%), which directly reflects the high utilization rate of transcriptome data. The clean reads of the sequencing data after quality control ranged from 43,615,300 to 55,692,524.

### 2.5. Differentially Expressed Genes (DEGs) in Pepper Peel Samples

The total clean readings were assembled into transcripts and compared with the reference gene model (including 41,727 genes). A total of 30,565 known genes and 2216 unknown genes were identified. There were 12,278 differential genes in the comparative expression of all treatments in 18 samples (deseq2 padj ≤ 0.05 |log2foldchange| ≥ = 0.0) (Figure 4).

### 2.6. Weighted Gene Co-Expression Network Analysis (WGCNA) Module Hierarchical Clustering Tree of DEGs in Pepper Peel Samples

To explore the key genes and co-expression networks that play important roles in the synthesis of anthocyanins in purple pepper pericarp, we analyzed 21,972 genes from 18 samples through WGCNA and constructed a cluster tree according to the correlation of expression between genes. Through the selection of the weighting coefficient, the power value corresponding to R^2^ is 0.8 was selected as the soft threshold. Figure 5 shows the hierarchical cluster tree of 25 modules of co-expressed genes analyzed by WGCNA, grouping genes with similar expression patterns into the same module. Genes in the same color block have similar expression trends and may be related by function.

### 2.7. Correlation Heatmap of Pepper Peel Samples and Modules and Kyoto Encyclopedia of Genes and Genomes (KEGG) Analysis of Sample-Specific Modules in Different Development Stages

To have a deeper understanding of the metabolic pathways that play an important role in the synthesis of anthocyanins in pepper pericarp, we performed a KEGG analysis on the genes on the above specific modules (*q* < 0.05) (Figure 6). The results showed that the “plant hormone signal transduction” pathway was significantly overexpressed in sample L168N2 (r = 0.89, *p* = 8× 10^−7^); “Ribosome”, “N-Glycan biosynthesis”, “protein export”, “protein processing in endoplasmic reticulum”, “toxic phosphorylation”, and “phagosome” pathways were significantly overexpressed in sample L72N3 (r = 0.74, *p* = 5 × 10^−4^); “Flavonoid biosynthesis”, “porphyrin and chlorophyll metabolism”, “photosynthesis antenna proteins”, “carbon metabolism”, “ubiquinone and other terpenoid-quinone biosynthesis”, “aminoacyl-tRNA biosynthesis”, “carbon fixation in photosynthetic organisms”, “photosynthesis”, “vitamin B6 metabolism”, “circadian rhythm–plant”, “pentose phosphate pathway”, and “fatty acid degradation” pathways were significantly overexpressed in sample L48N3 (r = 0.63, *p* = 0.005) (Table 3). It can be seen that the genes related to anthocyanin synthesis are mostly expressed in the early stage of light treatment (48 h).

### 2.8. Expression of Structural Genes in the Flavonoid Synthesis Pathway in Different Periods

The flavonoid pathway leads to the production of anthocyanins on the surface of pepper fruit, which is a key step affecting purple coloring. In the flavonoid metabolic pathway, we analyzed the transcriptome data of related enzymes. According to the changes in FPKM values of different genes in different treatments, *C4H* (107875406, 107875407), *CHI* (107871144, 107852750), and *CHS* (107864266, 107871256) genes had higher expression in pepper peel samples treated with light for 48 h, while the expression decreased gradually at 72 and 168 h. *DFR* (107860031), *F3’5’H* (107848667), *F3H* (107859880), and *LDOX* (107866341) genes were highly expressed in pepper peel samples treated with light for 48 h, 72 h, and 168 h, indicating that the expression of these late synthesis genes was relatively stable during the formation of flavonoids (Figure 7).

### 2.9. Mining of Anthocyanin-Related Light Response Genes

By analyzing the gene expression significance of the brown module, we obtained 305 genes with significant differential expression. These include genes related to anthocyanin biosynthesis, such as structural genes *CHS* (107871256107864266), *CHI* (107871144, 107852750), *DFR* (107860031), *F3’5’H* (107848667), *F3M* (107862334), *LDOX* (107866341), and *TCM* (107875406, 107875407). Some transporter genes, such as *GST* (107861273), *UDPGT* (107861697, 107843659), and *MATE* (10786323410784661), as well as transcription factors *EGL1* (107865400), *bHLH104* (107864591), and *WRKY44* (107843538, 107843524), were also overexpressed in this module. Refer to Appendix A for details of genes with significant differences in the brown module.

Among all the genes identified in this transcriptome, 19 related genes were overexpressed in the flavonoid biosynthesis pathway (sly00941). Refer to Table 4 for specific information on these genes. Most structural genes related to the flavonoid pathway were present in the brown module. This also confirmed that genes in the brown module may have the same function and expression pattern as anthocyanin synthesis. 

KEGG analysis of each module’s genes showed that the genes related to anthocyanin synthesis in pepper were mainly concentrated in the brown module. We used Cytoscape 37.1 to construct the co-expression network of the hub gene (107839364) in the brown module. As shown in Figure 8, we screened all genes with *q* < 0.05 and edge weight value ≥ 0.25 in the brown module, 91 associated genes were found closest to the hub gene 107839364 (Appendix A). Among them, *CHS*, *DFR*, *CHI*, and transcription factor *EGL1* have been identified as important genes involved in anthocyanin synthesis. The important roles of *MATE* transporters and *WRKY44* in anthocyanin synthesis and transport have also been reported. These genes have the same expression trend and may have similar functions. Therefore, it is speculated that these genes may be involved in anthocyanin biosynthesis in pepper peels.

### 2.10. Validation of Genes Related to Anthocyanin Synthesis

We selected important genes that may be related to anthocyanin synthesis for Quantitative Real-Time Polymerase Chain Reaction (qRT-PCR) verification, and the results are shown in Figure 9. It shows that the relative expression of each gene was related to FPKM, which confirmed that the gene expression measured by RNA-seq was reliable.

## 3. Discussion

### 3.1. WGCNA Revealed Key Genes Related to Anthocyanin Synthesis

Anthocyanin biosynthesis has a complex regulatory network. The synthetic pathway and metabolites of flavonoids have been the focus of anthocyanin biosynthesis in recent years. WGCNA was constructed using high-throughput sequencing to effectively classify these genes with the same expression pattern and function. WGCNA quickly screened the anthocyanin biosynthesis pathway related to light reaction and possible key regulators in other related pathways.

In this study, to explore the potential regulatory factors of anthocyanin biosynthesis in purple pepper peel, we studied the transcriptome data of 18 pepper peel samples and identified 305 genes with significant differences in the brown module, through WGCNA, including most structural genes of the anthocyanin synthesis. In the KEGG analysis of the brown module genes, 12 pathways including the flavonoid pathway were significantly enriched [32,33]. In the interaction network centered on 107839364 with the highest connectivity in the brown module, we also found the aggregation of genes of *CHS*, *DFR*, *CHI*, *EGL1*, *MATE*, *WRKY44*, and other genes, which provided further information for exploring the potential regulatory factors of anthocyanin biosynthesis in the peel of purple pepper. In our previous article, we also discussed the response of anthocyanin biosynthesis in pepper to light through strand-specific transcriptome and miRNA analysis [34]. The study showed that a large number of non-coding RNA in pepper peel may regulate the expression of anthocyanin synthesis structural genes after 48 h of light treatment.

### 3.2. Structural Genes Related to Anthocyanin Synthesis

The anthocyanin structural genes differentially expressed in the brown module mainly include *CHS* (107871256, 107864266), *CHI* (107871144, 107852750), *DFR* (107860031), *F3’5’H* (107848667), *F3M* (107862334), *LDOX* (107866341), and *TCM* (107875406, 107875407). Interestingly, all of these genes are significantly up-regulated (Table 2). It is worth noting that we found two significantly up-regulated *MATE* genes (107863234, 107844661). Studies have shown that MATE proteins have typical 12 transmembrane domains [35,36]. It is a transport molecule of cation reverse transport, including hormones, organic acids, and other secondary metabolites. Na^+^ or H^+^ ions exchanged across the membrane are the driving force of substrate transport [37]. The MATE transporter of *Arabidopsis* is involved in the transport of proanthocyanidin biosynthetic precursors in the seed coat, which may control the vacuolar isolation of flavonoids in the seed coat endothelium of *Arabidopsis* [8,38]. Therefore, *MATE* (107863234, 107844661) genes may play an important role in the transport of anthocyanins in the peel of purple pepper.

### 3.3. Regulatory Genes Related to Anthocyanin Synthesis

*MYB1R1* (107840959) is the only MYB gene differentially expressed in the brown module. The MYB family is the most important transcription factor family regulating anthocyanin synthesis, and MYB1R1 is also R2R3 MYB, indicating that it may play an important role in the synthesis of anthocyanins in purple pepper [39,40].

F3’H and F3’5’H are important branching points in the flavonoid synthesis pathway. WRKY44 can bind to the promoters of these two genes to regulate the accumulation of specified anthocyanin types [41,42]. WRKY44 regulates catechin branching pathway synthesis in the flavonoid pathway [3,43]. WRKY44 is also considered to regulate the synthesis of anthocyanins in pitaya fruit pulp and is identified as the central gene in the interaction gene network constructed by light response anthocyanin synthesis to regulate the expression of anthocyanin structural genes [44,45]. Two *WRKY44* (107843538, 107843524) genes found in the hub gene interaction network also showed significant up-regulation. This provides a new idea for us to study the key regulatory genes of anthocyanin synthesis.

We also found three *bHLH* genes that were significantly up-regulated in the genes with significant differences in the brown module, namely *EGL1* (107865400), *bHLH**104* (107864591), and *HLH* (107842687). MdbHLH3 can combine with the promoters of anthocyanin biosynthesis genes DFR and UFGT and the promoters of regulatory gene MdMYB1 to activate their expression [46]. Under UV-AB treatment, the structural genes *F3’5’H*, *DFR*, *ANS*, and *UFGT*, as well as the regulatory genes *EGL1* and *TT2* of purple tea showed up-regulated expression [47]. Therefore, *EGL1* may play an important role in anthocyanin synthesis.

The combined transport of glutathione S-transferase (GST) and flavonoids can improve the ability to protect plant cells from salt stress [9,48]. *GST* is very important for the coloring of red flesh peach. Virus-induced *PpGST1* gene silencing in red meat peach not only reduced anthocyanin accumulation but also decreased the expression of anthocyanin biosynthesis and regulatory genes [7]. Cyclamen-containing anthocyanins in petals turn white after the deletion of the *GST1* allele, indicating that *GST1* is a gene for anthocyanin transport in Cyclamen [49]. The up-regulated expression of *GST* (107861273), a significantly different gene in the brown module, also shows that it plays an important role in the anthocyanin synthesis of purple pepper fruit.

## 4. Materials and Methods

### 4.1. Material Preparation

The purple pepper germplasm HNUCA21 (*Capsicum annuum*) obtained from the germplasm bank of Hainan University was uniformly placed under controlled drip irrigation equipment for management, and Stanley 20-19-19 fourth element water-soluble fertilizer was applied for the fertilizer. When the pepper entered the full flowering period, all the flowers that opened on the same day were marked. On the third day after marking, all fruits that had just begun to expand were bagged and protected from light. After two weeks of bagging and the development of pepper fruits, 20 well-developed fruits on the best single plant were selected on a sunny morning. After removing the covering on one side of the fruit, they were cultured under natural light for 48 h, 72 h, and 168 h. Six fruits were randomly collected from the upper, middle, and lower parts of the plant each time. After removing the bag, fruits were cut in two along the covering line and the position of the fruit pedicle and fruit top by 1 cm, respectively. Additionally, 0.5 cm along the edge of the covering line was also cut off. The processed samples were marked and wrapped with tin foil, then put into liquid nitrogen for thorough freezing and stored in a refrigerator at −80 °C. The samples were used for subsequent metabolite extraction, RNA extraction, and RNA-seq. The samples of light and dark in the three periods were named L48, D48, L72, D72, L168, and D168, respectively.

### 4.2. Measurement of Total Anthocyanin, Chlorophyll, and Carotenoid Concentrations

After grinding, the 0.2 g sample was immersed in 1% acetic acid methanol solution and kept in the dark at 4 °C overnight. Centrifugation was carried out at 15,000× *g* rpm for 3 min, after adding the same amount of supernatant to the buffer solution with a pH of 1.0 and 4.5, respectively. The spectrophotometer (DU800) was used to measure the absorbance value at 510 and 700 nm. A 1% acetic acid methanol solution was used as a control and 1 g of peel was thoroughly ground and soaked in 6 mL of 80% cold acetone, then centrifuged at 4 °C and 15,000× *g* rpm for 10 min. The absorbance values were determined at 440, 645, and 663 nm using a spectrophotometer. Overall, 80% acetone was used as a control. Refer to Bai et al. [2] for details.

### 4.3. Determination of Flavonoids by LC-ESI-MS/MS

Ultra-performance liquid chromatography (UPLC) (ExionLC™AD, https://sciex.com.cn/, accessed on 24 July 2022) with tandem mass spectrometry (MS/MS) (QTRAP^®^6500+, http://sciex.com.cn/, accessed on 24 July 2022) (LC-ESI-MS/MS) was used for metabolite analysis on 05 August 2021 [50]. A freeze-dried sample was ground with zirconium beads, 50 mg of sample powder at 500 μL extract from the extract (50% methanol aqueous solution of 0.1% hydrochloric acid) was added, and then centrifuged to extract the supernatant. Finally, the supernatant was filtered by a microporous membrane (pore size 0.22 μm).

Liquid phase condition: chromatographic column: ACQUITY BEH C18 1.7 µm, 2.1 mm × 100 mm; mobile phase: phase A was ultrapure water (add 0.1% formic acid), and phase B was methanol (add 0.1% formic acid); elution gradient: the proportion of phase B was 5% at 0.00 min, increased to 50% at 6.00 min, increased to 95% at 12 min, maintained for 2 min, decreased to 5% at 14 min, and the equilibrium time was 2 min; the flow rate was 0.35 mL/min; column temperature was 40 °C; and the injection volume was 2 μL.

Mass spectrometry conditions: electron ionization temperature was 550 °C and positive ion mass spectrometry voltage was 5500 .V Curtain gas was pressurized as 35 psi. Scanning and detection were conducted according to declustering potential and collision energy.

The database was built based on the standard sample to qualitatively analyze the data detected by mass spectrometry. Quantitative analysis was completed by the multiple reaction monitoring of triple quadrupole mass spectrometry.

### 4.4. RNA-Seq

The samples were sent to Novogene Bioinformatics Technology Co. Ltd. (Beijing, China) for transcriptome sequencing. The total RNA in the sample was extracted with Trizol Reagent (Invitrogen, Carlsbad, CA, USA). After the RNA purity (OD260/280 and OD260/230 ratio) was detected by nanophotometer spectrophotometer and the RNA integrity was accurately detected by Agilent 2100 Bioanalyzer, the database was established. The library building kit used in the library building was NEBNext of Illumina^®^ UltraTM RNA Library Prep Kit. The mRNA with polyA tail was enriched by Oligo (T) magnetic beads, and the second strand of cDNA was synthesized from dNTPs. After end repair, adding A-tail and connecting sequencing connector, the cDNA of about 250~300 bp was screened and amplified by PCR, and finally, the library was obtained. Removed low-quality reads (reads with a base number of Qphred ≤ 20 accounting for more than 50% of the whole read length). Meanwhile, Q20, Q30, and GC contents of clean data were calculated. The sequencing platform was Illumina Hiseq TM2000, and the sample had three biological replicates. Reference genome downloaded from https://www.ncbi.nlm.nih.gov/genome/10896 on 2 December 2021.

### 4.5. Differential Expression Analysis

DE mRNA was determined by the Ballground method. The differential expression analysis between the two comparison combinations was performed using DEseq2 software (1.16.1). Genes with *p* < 0.05 were considered differentially expressed genes. The corrected *p*-value and |log2foldchange| were used as the threshold of significant differential expression. KEGG (Kyoto Encyclopedia of Genes and Genomes) is the main public database related to this pathway (https://www.genome.jp/kegg/, accessed on 24 July 2022).

### 4.6. RNA Extraction and Quantitative Reverse Transcription PCR

Total RNA was extracted with RNA prep pure plant plus Kit (Tian Gen, Beijing, China). According to the manufacturer’s protocol, RT Super Mix (Tian Gen, Beijing, China) was eliminated with Fast King gDNA to complete the cDNA synthesis process. Specific primers were designed using Primer 5. The expression of important unigenes in the metabolic pathway was confirmed by the Light Cycle 96 Real-Time PCR system (Roche, Switzerland) [2]. Based on 2^−ΔΔCT^ method was used to evaluate the relative expression level of the target gene. See Appendix A for primer information.

### 4.7. Statistical Analysis

The data processing system (DPS) version 7.05 software was used. The heatmaps were drawn by TBtools software. R-package WGCNA was used to calculate the functional set, network construction, gene screening, etc. of weighted association analysis. Cytoscape 3.7.1 software was used to construct the co-expression network of genes [4].

## 5. Conclusions

Anthocyanins in purple pepper fruit only accumulated in a small amount in the pericarp after 48 h of light treatment, but a large number of related genes affecting anthocyanin synthesis have been expressed rapidly. Ten anthocyanin synthesis structural genes, three transporters, and six important transcription factor genes belong to the brown module, and their expression is significantly up-regulated, indicating that the genes of this module may play a common regulatory role in anthocyanin synthesis. In particular, *CHS*, *DFR*, *CHI*, *EGL1*, *MATE*, and *WRKY44* are highly connected with the hub gene 107839364 in the gene co-expression network. Our study further supports the hypothesis that vacuolar isolation of flavonoids involves vesicle transport, membrane transporter, and glutathione S-transferase. 

## Figures and Tables

**Figure 1 ijms-23-08357-f001:**
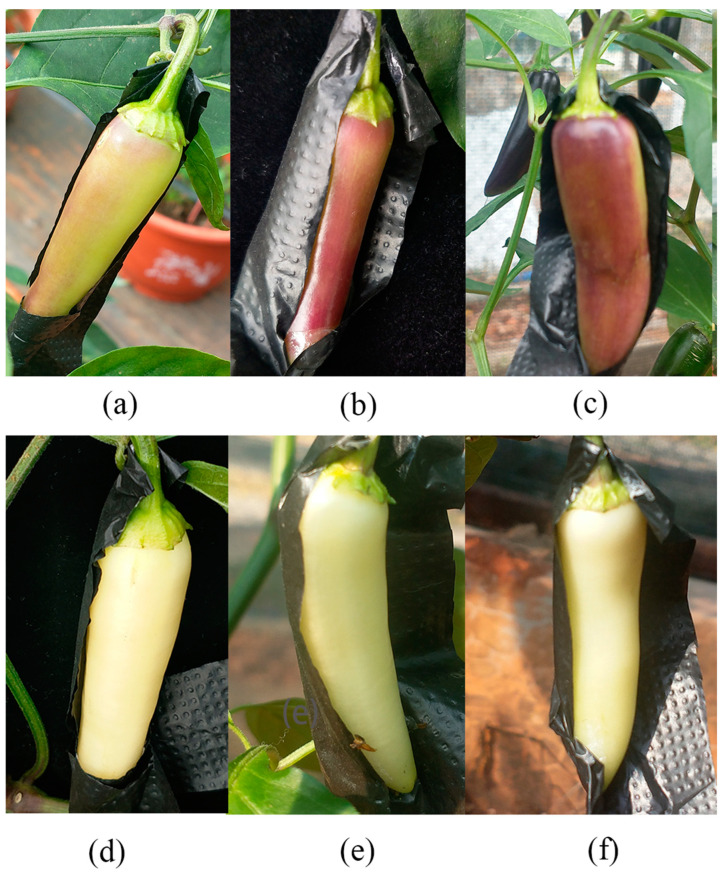
Changes in appearance color of purple pepper fruit under different light treatments. The coloring of pepper fruit color after light treatment (**a**–**c**) and shaded treatment (**d**–**f**) for 48 h, 72 h, and 168 h.

**Figure 2 ijms-23-08357-f002:**
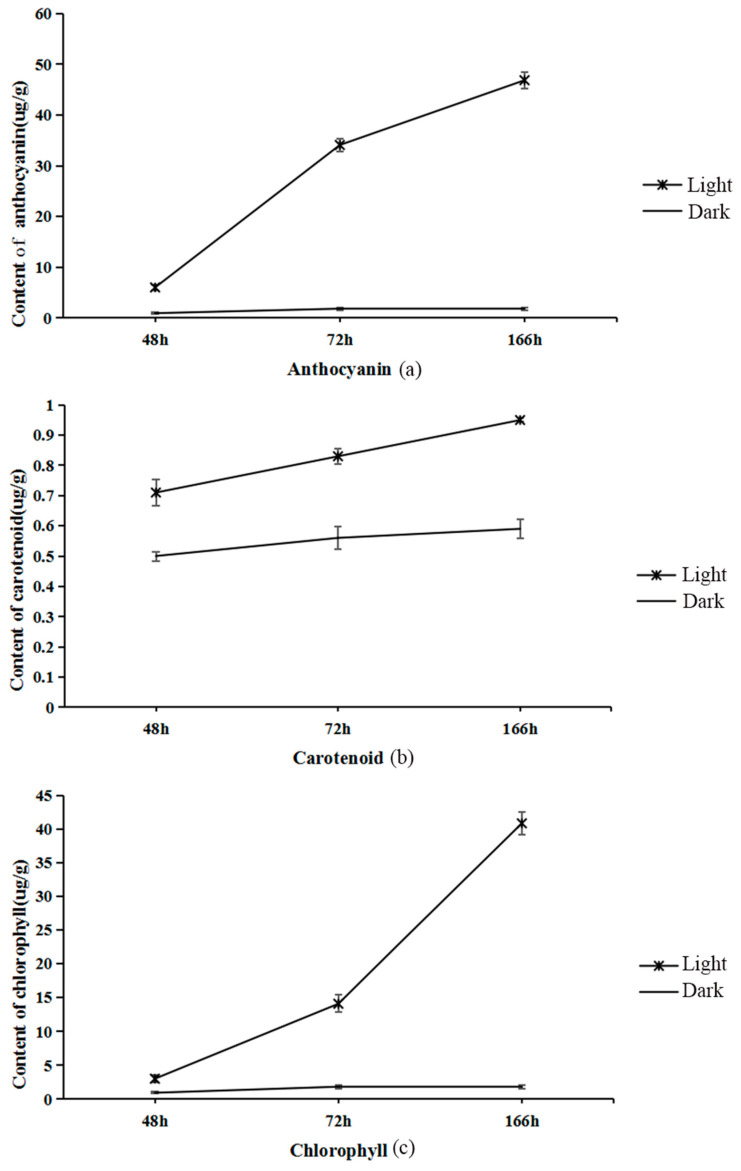
The content of anthocyanins, carotenoids, and chlorophyll in the peel of capsicum. The x-axes are the times of different treatments, and the y-axes represent the content of various substances, anthocyanins (**a**), carotenoids (**b**), and chlorophyll (**c**). Unit µg/g freeze-dried). The pictures show the average value of three biological repetitions. The error bars indicate the standard deviation.

**Figure 3 ijms-23-08357-f003:**
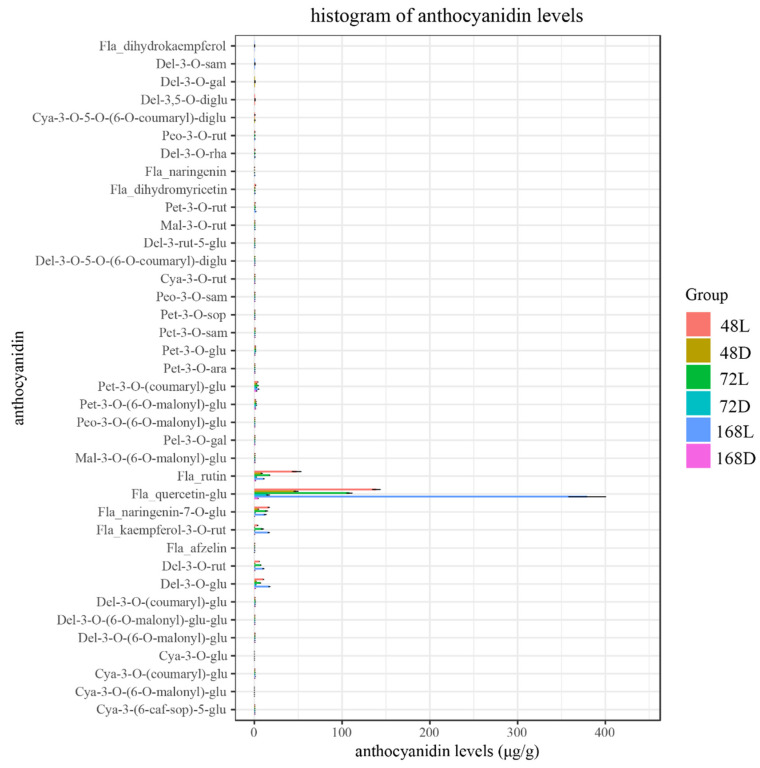
Histogram of the flavonoid type and content in each sample. The abscissa is content. The ordinate is the type. The picture shows the average value of three biological repetitions. The error bars indicate the standard errors.

**Figure 4 ijms-23-08357-f004:**
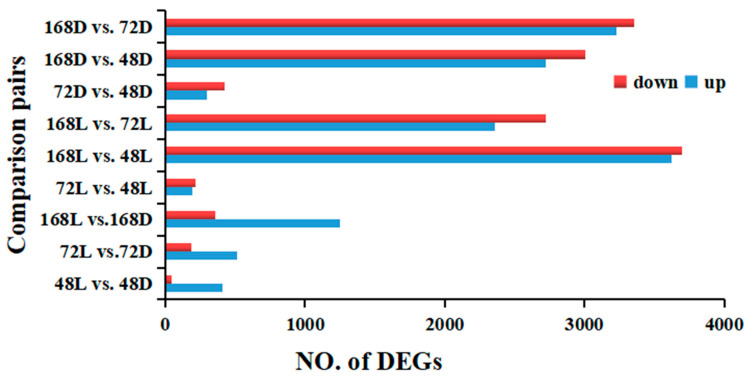
Number of DEGs in samples treated with different light treatments on the peel of peppers.

**Figure 5 ijms-23-08357-f005:**
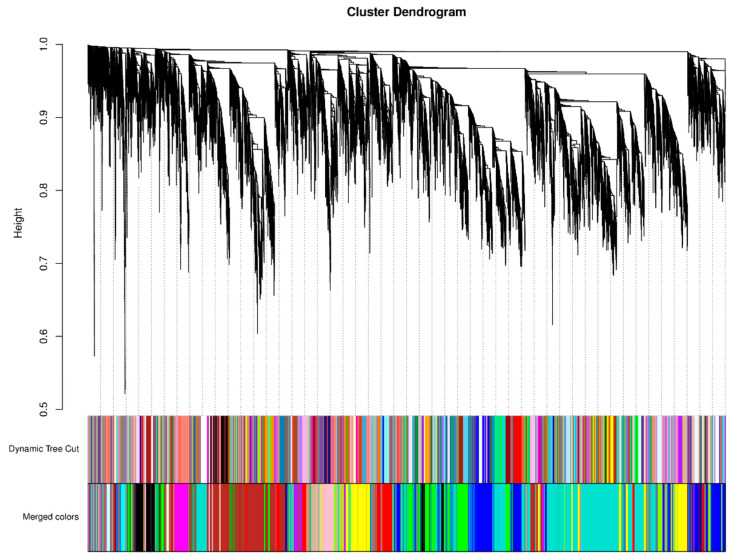
Hierarchical clustering tree for identifying gene co-expression modules by clustering. Dynamic Tree Cut is to obtain different module diagrams by using the dynamic cutting method, in which different colors represent different modules. Merged colors at the bottom are the combined diagram of modules with a dissimilarity coefficient of less than 0.25, in which different colors represent the combined module tree view. The longitudinal distance represents the distance between two nodes (between genes), and the transverse distance is meaningless.

**Figure 6 ijms-23-08357-f006:**
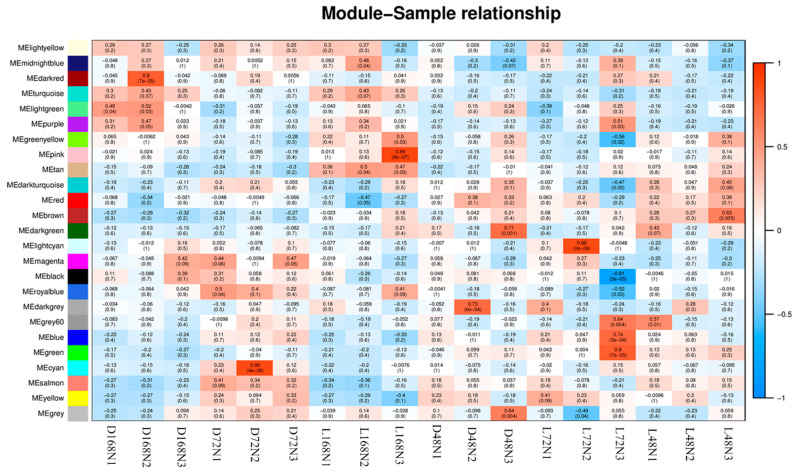
Heatmap of the sample and inter-module correlation. The abscissa is the sample and the ordinate is the module. The number of each grid represents the correlation between the module and the sample. The closer the value is to 1, the stronger the positive correlation between the module and the sample; The closer to −1, the stronger the negative correlation between the module and the sample. The numbers in brackets represent the significance *p*-value. The smaller the value, the stronger the significance.

**Figure 7 ijms-23-08357-f007:**
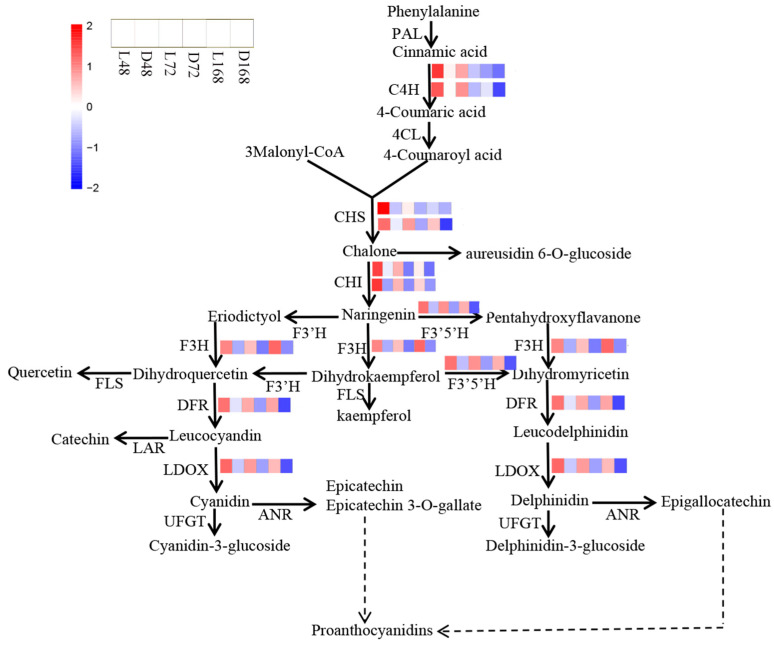
Expression patterns of DEGs in flavonoid synthesis pathways of different light treatment samples. The heatmap shows the expression patterns of differentially expressed structural genes related to the flavonoid pathway in pepper peels at different stages (upper left). Each row represents a unigene, and the expression change in the unigenes is based on the FPKM (the number of kilobase fragments of exon per million reads). The closer to blue, the lower the expression, and the closer to red, the higher the expression.

**Figure 8 ijms-23-08357-f008:**
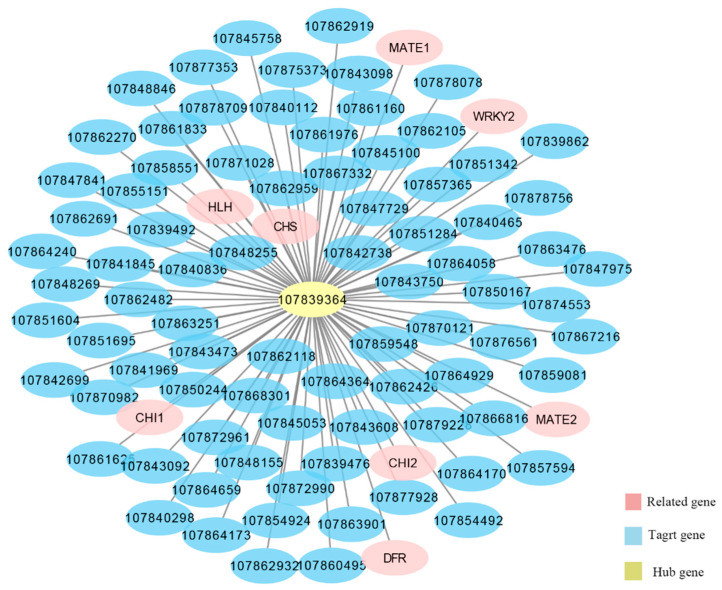
Co-expression network analysis of the hub genes. Take 107839364 with the best connectivity in the brown module as the central gene, and the network structure diagram of all genes with a significant difference (*q* < 0.05) and edge weight ≥ 0.25 in the brown module.

**Figure 9 ijms-23-08357-f009:**
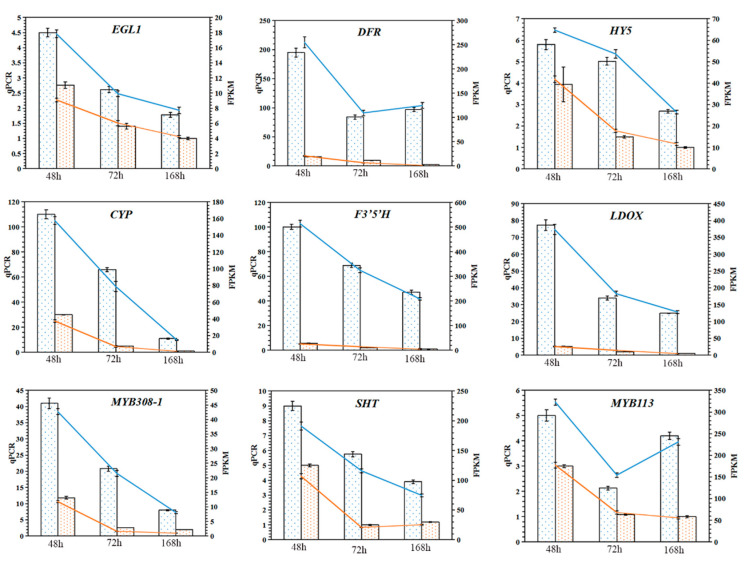
qPCR validation of genes involved in anthocyanin synthesis. The expression profiles of 9 genes were measured by RNA-seq and qRT PCR. These nine genes include structural genes and transcription factors related to the flavonoid pathway. The histogram data represent qRT PCR data (left *y*-axis) and the broken line represents RNA-seq data (right *y*-axis). Blue dots in the histogram represent light treatment and orange dots represent dark treatment. The *x*-axis in each chart represents three-light treatment stages (48 h, 72 h, and 168 h). The qRT PCR assay had three biological replicates, each with three technical replicates (*n* = 9). The FPKM value of RNA-seq has three biological repetitions (*n* = 3). The error bars indicate the standard deviation.

**Table 1 ijms-23-08357-t001:** Quality evaluation of sample sequencing output data for strand-specific RNA libraries.

Sample	Clean_Reads	Clean_Bases	Q30	GC_pct
L48N1	46,636,752	7.0 G	94.36	42.57
L48N2	46,049,962	6.91 G	94.52	42.46
L48N3	46,999,312	7.05 G	94.41	42.43
D48N1	54,821,486	8.22 G	94.01	42.33
D48N2	42,996,712	6.45 G	94.39	42.5
D48N3	44,649,882	6.7 G	94.28	42.39
L72N1	48,664,980	7.3 G	94.08	42.44
L72N2	45,198,608	6.78 G	94.37	42.39
L72N3	46,525,636	6.98 G	94.27	42.97
D72N1	52,924,474	7.94 G	94.28	42.19
D72N2	44,877,252	6.73 G	94.18	42.42
D72N3	46,212,326	6.93 G	94.36	42.43
L168N1	46,145,536	6.92 G	94.11	42.06
L168N2	44,256,704	6.64 G	94.43	41.88
L168N3	45,263,740	6.79 G	94.19	41.98
D168N1	46,224,450	6.93 G	94.48	42.15
D168N2	46,802,178	7.02 G	94.56	42.1
D168N3	44,491,876	6.67 G	94.11	41.74

Note: clean reads: total number of pair-end reads in clean data. Clean bases: total base of clean data; GC_pct: GC content of clean data, that is, the percentage of G and C bases in total bases in clean data; Q30: percentage of bases with a clean data quality value greater than or equal to 30.

**Table 2 ijms-23-08357-t002:** Reads of strand-specific RNA libraries and reference genome comparison list.

Sample	Total_Reads	Total_Map(%)	Unique_Map(%)	Multi_Map(%)	Positive_Map(%)	Negative_Map(%)
L48N1	46,636,752	95.58%	92.33%	3.25%	46.13%	46.20%
L48N2	46,049,962	83.78%	80.94%	2.84%	40.45%	40.50%
L48N3	46,999,312	95.64%	92.23%	3.40%	46.08%	46.15%
D48N1	54,821,486	89.29%	86.47%	2.82%	43.21%	43.26%
D48N2	42,996,712	84.37%	81.66%	2.71%	40.80%	40.86%
D48N3	44,649,882	95.31%	92.10%	3.20%	46.02%	46.08%
L72N1	48,664,980	94.67%	91.60%	3.07%	45.77%	45.83%
L72N2	45,198,608	75.08%	72.65%	2.43%	36.31%	36.34%
L72N3	46,525,636	96.09%	92.97%	3.11%	46.44%	46.53%
D72N1	52,924,474	95.01%	92.08%	2.93%	46.01%	46.06%
D72N2	44,877,252	95.53%	92.64%	2.89%	46.29%	46.35%
D72N3	46,212,326	95.59%	92.61%	2.98%	46.28%	46.33%
L168N1	46,145,536	87.66%	84.78%	2.88%	42.36%	42.41%
L168N2	44,256,704	80.60%	77.95%	2.65%	38.96%	38.99%
L168N3	45,263,740	95.02%	92.00%	3.02%	45.99%	46.01%
D168N1	46,224,450	85.77%	83.19%	2.57%	41.58%	41.62%
D168N2	46,802,178	77.71%	75.31%	2.40%	37.64%	37.67%
D168N3	44,491,876	78.57%	76.26%	2.31%	38.12%	38.15%

Note: total_ reads: the number of clean reads of sequencing data after quality control; total_ map (%): percentage of reads compared to the genome; unique_ map (%): percentage of reads compared to the unique location of the reference genome; multi_ map (%): percentage of reads compared to multiple locations of the reference genome; positive_ map (%): percentage of reads compared to the positive chain of the reference genome; negative_ map (%): percentage of reads compared to the negative chain of the reference genome.

**Table 3 ijms-23-08357-t003:** KEGG analysis of sample-specific modules for different light treatments.

Module	Sample	KEGGID	Description	GeneRatio	padj
pink	L168N2	sly04075	Plant hormone signal transduction	16/88	2.64 × 10^−4^
blue	L72N3	sly03010	Ribosome	72/526	5.36 × 10^−12^
		sly03050	Proteasome	18/526	1.39 × 10^−4^
		sly00510	N-Glycan biosynthesis	14/526	1.23 × 10^−2^
		sly03060	Protein export	13/526	2.02 × 10^−2^
		sly04141	Protein processing in endoplasmic reticulum	37/526	3.93 × 10^−2^
		sly00190	Oxidative phosphorylation	25/526	4.02 × 10^−2^
		sly04145	Phagosome	18/526	4.02 × 10^−2^
brown	L48N3	sly00860	Porphyrin and chlorophyll metabolism	19/487	1.63 × 10^−5^
		sly00196	Photosynthesis antenna proteins	11/487	4.48 × 10^−5^
		sly01200	Carbon metabolism	52/487	1.10 × 10^−4^
		sly00130	Ubiquinone and other terpenoid-quinone biosynthesis	14/487	1.28 × 10^−3^
		sly00970	Aminoacyl-tRNA biosynthesis	15/487	1.87 × 10^−3^
		sly00710	Carbon fixation in photosynthetic organisms	19/487	3.31 × 10^−3^
		sly00195	Photosynthesis	18/487	1.03 × 10^−2^
		sly00941	Flavonoid biosynthesis	12/487	1.44 × 10^−2^
		sly00750	Vitamin B6 metabolism	6/487	1.80 × 10^−2^
		sly04712	Circadian rhythm—plant	11/487	2.57 × 10^−2^
		sly00030	Pentose phosphate pathway	13/487	2.63 × 10^−2^
		sly00071	Fatty acid degradation	13/487	2.90 × 10^−2^

**Table 4 ijms-23-08357-t004:** Flavonoid biosynthesis pathway gene annotation.

Gene_ID	Module	Gene_Description
107875625	Blue	agmatine coumaroyltransferase-2-like%2C transcript variant X2
107850816	Blue	omega-hydroxypalmitate O-feruloyl transferase
107851884	Brown	acetyl-CoA-benzylalcohol acetyltransferase-like
107848097	Brown	agmatine coumaroyltransferase-2-like
107843659	Brown	anthocyanidin 3-O-glucosyltransferase-like
107864266	Brown	chalcone synthase 1B
107871256	Brown	chalcone synthase 2
107852750	brown	chalcone--flavonone isomerase%2C transcript variant X1
107844023	brown	cytochrome P450 98A2-like
107860031	brown	dihydroflavonol-4-reductase
107862334	brown	flavonoid 3′-monooxygenase
107866341	brown	leucoanthocyanidin dioxygenase
107871144	brown	probable chalcone--flavonone isomerase 3%2C transcript variant X2
107840863	brown	shikimate O-hydroxycinnamoyltransferase-like
107860278	green	caffeoyl-CoA O-methyltransferase%2C transcript variant X2
107853336	green	probable caffeoyl-CoA O-methyltransferase At4g26220%2C transcript variant X1
107840262	magenta	caffeoyl-CoA O-methyltransferase 6
107840069	turquoise	acetyl-CoA-benzylalcohol acetyltransferase-like
107839366	turquoise	cytochrome P450 98A2-like

## Data Availability

Supporting reported data can be found at https://www.ncbi.nlm.nih.gov/sra/PRJNA846152 from 26 July 2022.

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
