# Peer review of "Response of Anthocyanin Accumulation in Pepper (Capsicum annuum) Fruit to Light Days"

_ijms, 2022, doi:10.3390/ijms23158357_

Round 1
Reviewer 1 Report
Dear editor and authors!
The article takes into consideration an interesting problem of response of anthocyanin accumulation to light. Even despite the rich hoard of literature on this topic, new discoveries are made daily, which makes it a highly current issue. It is especially so that the plant being investigated is a crop plant and the compounds synthesis of is the main focus of investigation are crucially important for the quality of the crop.
I find the article very interesting and I think it will be to the liking of readers of IJMS, there are however some minor improvements that would have to be included before publication.
Minor Revision
Comments:
- Editing by a native speaker would be beneficial. The work is understandably written and the linguistic side is not very coarse, but language editing would have been a boon to the article.
-All abreviations should be explained upon first appearance in the text
-Line 22: WGCNA – provide full name, not everyone has to know it stands for weighted gene co-expression network analysis
-Line 34: “…genes in anthocyanin…”
-Lines 76-77: “energy plant” is quite informal. Maybe mitochondria?
-Lines 80-81: The sentence is unclear. “…the colour is the deepest then.”?
-Line 100: Loads is not quite formal. Maybe abundance…? Some synonym?
-Lines 105, 109,113 and many more in the work: it is a shaded surface, not shading surface. Otherwise, the fruit would have to do the shading with the investigated surface.
Line 109: Again, change “greatly” to “significantly”. It is too informal.
Lines 135-139: Edit the names of the compounds to proper chemical names. I understand the “Fla” prefix is added by the software, but it needs to be replaced by proper chemical names.
Line 142: Why standard errors? Were the actual errors not calculated? This needs to be explained.
Line 163: “…compared to…”
Line 186: Explain DEGs- first time in text
Line 199: “…trends, and may be related by function”
Line 215 and following text: “enriched” is highly informal and unfortunate. Please do find a more formal expression. Maybe simply “overexpressed”?
Line 260: “Synthetic” would mean artificial. “…these genes responsible for synthesis which were expressed late in the experiment.” For example.
Lines 276-278: Entire sentence is unclear and needs to be rewritten. “…overexpressed in the flavonoid…”
Line 308: FPKM- meaning of the abbreviation?
Line 332: “…genes…”
Lines 396-398: Sentence is unclear and needs to be rewritten. Does the author mean that the allele leads to white flowers?
Line 419: “…cut in two…”
Entire Matherials and Methods section has to be rewritten in Passive. As is, the grammatical tense switches back and forth. it is unacceptable.
Lines 442-465: All well and good, but how were the compounds identified? By comparison to standards run on the same system? Ion fragment databases? It has to be described, lack of this information putts in doubt entire biochemical part of this work.
Lines 518-519: “…only in a small amount…”
Reviewer 2 Report
The aim of this current study is to investigate anthocyanin accumulation and identify changes in gene expression in pepper fruit depending on light during ripening. I have some minor suggestions to authors:
Clear aim of study must be formulated at the end of introduction.
Materials and methods must be rewritten in past tense.
Statistics should be described in figure names (standard error, p level, etc.).
Self-citation in conclusions is note very common, it could be moved to discussion. Conclusions should focus on current study.
In general, I think manuscript may be published after minor revision.
